# Optimizing the Conditions of Cationic Polyacrylamide Inverse Emulsion Synthesis Reaction to Obtain High–Molecular–Weight Polymers

**DOI:** 10.3390/polym14142866

**Published:** 2022-07-14

**Authors:** Tung Huy Nguyen, Nhung Thi Nguyen, Thao Thi Phuong Nguyen, Ngoc Thi Doan, Lam Anh Thi Tran, Linh Pham Duy Nguyen, Thanh Tien Bui

**Affiliations:** Center for Polymer Composite and Paper, School of Chemical Engineering, Hanoi University of Science and Technology, Hai Ba Trung District, Hanoi 11600, Vietnam; tung.nguyenhuy@hust.edu.vn (T.H.N.); nhung.nguyenthi@atpcorporation.com.vn (N.T.N.); thao.nguyenthiphuong@atpcorporation.com.vn (T.T.P.N.); ngoc.doanthi@atpcorporation.com.vn (N.T.D.); lamanh.tranthi@atpcorporation.com.vn (L.A.T.T.)

**Keywords:** CPAM, copolymer cation, box Behnken, response surface methodology, experimental plan

## Abstract

Cationic polyacrylamide (CPAM) emulsifier is widely applied in the wastewater treatment industry, mining industry, paper industry, cosmetic chemistry, etc. However, optimization of input parameters in the synthesis of CPAM by using the traditional approach (i.e., changing one factor while leaving the others fixed at a particular set of conditions) would require a long time and a high cost of input materials. Onsite mass production of CPAM requires fast optimization of input parameters (i.e., stirring speed, reaction temperature and time, the amount of initiator, etc.) to minimize the production cost of specific–molecular–weight CPAM. Therefore, in this study, we synthesized CPAM using reverse emulsion copolymerization, and proposed response surface models for predicting the average molecular weight and reaction yield based on those input parameters. This study offers a time–saving tool for onsite mass production of specific–molecular–weight CPAM. Based on our response surface models, we obtained the optimal conditions for the synthesis of CPAM emulsions, which yielded medium–molecular–weight polymers and high conversion, with a reaction temperature of 60–62 °C, stirring speed of 2500–2600 rpm, and reaction time of 7 h. Quadratic models showed a good fit for predicting molecular weight (Adj.R^2^ = 0.9888, coefficient of variation = 2.08%) and reaction yield (Adj.R^2^ = 0.9982, coefficient of variation = 0.50%). The models suggested by our study would benefit the cost–minimization of CPAM mass production, where one could find optimal conditions for synthesizing different molecular weights of CPAM more quickly than via the traditional approach.

## 1. Introduction

Industries produce wastewater, including various components such as suspended solids, organic and inorganic particles, dissolved ions, etc. Sewage treatment is essential for environmental safety and sustainable development. One of the most significant industrial procedures for wastewater treatment is flocculation, which is the process of aggregation of particles caused by chemical coagulants. Flocculation is extensively used due to its ease of use, high efficiency, and cost savings [1]. Among popularly used chemical coagulants, high–molecular–weight synthetic polymers have been widely employed as flocculants in colloidal suspensions to separate and dewater solid/water systems [2,3]. Polyacrylamide, a water–soluble polymer formed by the polymerization of acrylamide monomers, is among the most used chemicals for wastewater treatment and sludge dewatering [4,5,6]. Cationic polyacrylamide (CPAM) is one of the most widely applied polymers due to its high performance in flocculation, sludge dewatering, and harvesting microalgae [6,7,8,9,10,11,12,13].

There are many studies on CPAM synthesis technology, such as the free radical polymerization method, grafting method, and polymer modification, in which modified polymers often have sizeable molecular weight [10,11,14,15,16,17,18,19,20,21,22,23]. CPAM synthesized by the grafting method is biodegradable but not stable; it has a short storage time and has low molecular weight [24]. Free radical polymerization by inverse emulsion is considered a better method to generate CPAM with a high molecular weight, faster reaction rate, high conversion efficiency, and ease of controlling the temperature of the reaction [12,25]. In studying the fusion reaction of water–in–oil (W/O) emulsions of CPAM, Barari et al. showed the influence of factors such as stirring speed, reaction time, reaction temperature, initiator content, and emulsifier content on the average molecular weight of the obtained polymer, as well as the conversion efficiency of the reaction [26]. Mohsin and Attia synthesized polyacrylamide emulsions to stabilize dunes in arid regions by reacting water–in–oil emulsions at a stirring speed of 2000–3000 rpm, with reaction temperature of 50–60 °C [27]. Although previous researchers have studied the effects of input parameters on the molecular weight and conversion efficiency of CPAM, none has suggested a relationship between input parameters and molecular weight/conversion efficiency. Such a relationship would shorten the time for finding the optimal conditions of CPAM production via simple calculations of input parameters (i.e., temperature, stirring speed, and reaction time). Such a method would be quicker than the traditional approach (i.e., changing one factor while leaving the others fixed at a particular set of conditions), which requires a long time and a high cost of input materials.

Additionally, due to the short storage time and high demand for CPAM, there is a need for fast optimization of synthesis conditions for onsite mass production of CPAM. Companies and factories could minimize production costs by reducing the time required to obtain optimized conditions for producing a specific–molecular–weight CPAM. Response surface methodology (RSM) allows the solution of multivariable equations and evaluation of the relative significance of several relevant factors even in the presence of complex interactions [28,29,30]. RSM has been widely used for optimizing environmental processes such as physicochemical removal of dyes from wastewater and the coagulation–flocculation process [28,31]. However, to the best of our knowledge, to date, no studies have applied response surface methodology for the production of CPAM.

This study aims to develop a time–saving tool for onsite mass production of specific–molecular–weight CPAM. For that purpose, we synthesized CPAM using reverse emulsion copolymerization, and proposed response surface models containing three parameters (i.e., temperature, stirring speed, and reaction time) to predict CPAM’s molecular weight and conversion efficiency. Hydrogen magnetic resonance spectroscopy (^1^H–NMR) and Fourier–transform infrared (FTIR) spectroscopy were used to confirm the structure of the polymers. Gel permeation chromatography (GPC) and a viscosity meter were used to confirm the molecular weight and molecular weight distribution of the desired CPAM. Dynamic light scattering (DLS) was used to confirm the particle size distribution of CPAM.

## 2. Materials and Methods

### 2.1. Chemicals

Industrial monomer acrylamide (AM, 98%) was purchased from Jinjinle Chemical Co., Ltd. (Zhuhai, China). Methyl acrylacyl oxyethyl trimethyl ammonium chloride (DMC) was bought from Wuxi Xinyu Chemical Co., Ltd. (Yixing, Jiangsu, China) as an aqueous solution (74.68%). Isopar L (Exxon), vegetable oil (EFKO Russian), Span 80 (99.5%), and Tween 85 (99.5%) were purchased from Beijing Chemical Reagent Company (Beijing, China). 2,2′–Azobis(2–methylpropionamidine) dihydrochloride (V50, 98.0%) and azobisisobutyronitril (V60, 98%) were purchased from Tokyo Chemical Industry Co., Ltd. IPA (99%), I_2_ (≥99%), ethanol (≥99.5%), KI (≥99%), HgCl_2_ (≥99.5%), Na_2_S_2_O_3_.5H_2_O (≥99%), and soluble starch (≥99%) were supplied by Xylong, and other chemicals (analytically pure) were used without further purification.

### 2.2. Preparation of CPAM (W/O Emulsions)

CPAM was prepared by reverse emulsion copolymerization via the free radical mechanism, with AM:DMC ratio of 1:4 (Figure 1). The protocol of Liu et al. was followed, with some modifications [32]. Each reaction was conducted in a 500 mL, three–necked, round–bottomed flask equipped with a mechanical stirrer, a thermometer, a glass spigot, and a high–purity nitrogen inlet/outlet (Figure 2).

Nitrogen gas was continuously aerated during the reaction. UV lamp and a photoinitiator were used to start phase 1 of reaction, and then after phase 2 of the reaction, with stirring and slowly adding the redox initiator system, the factors affecting the polymerization reaction—stirring speed, reaction time, and temperature—were investigated according to the experimental matrix table. An inverse emulsion cationic polymer was formed at the end of the process following the synthesis reaction shown in Figure 3.

### 2.3. Determination of the Conversion and Molecular Weight of CPAM

The W/O emulsion product was converted to O/W by using NP–10. Then, the O/W emulsions were put into a 100 mL beaker, and isopropyl alcohol was slowly added until the solution was clear. The precipitation was filtered and dried at 45 °C to constant mass for molecular mass determination using a vacuum oven cabinet, and “filtrate X” was used for titration and conversion.

The overall monomer conversion was determined by the residual content of the participating monomers (AM and DMC), using the HIP method. HIP solution ($I_{2}$ and ${\mathrm{HgCl}}_{2}$ in ethanol) was added to “filtrate X”, and $I_{2}$ reacted with $\;{\mathrm{HgCl}}_{2}$ to produce ICl, which was added to the excess AM and DMC in “filtrate X”. We used KI to reduce the excess ICl to obtain $I_{2}$. By titration of $I_{2}$ with ${\mathrm{Na}}_{2}S_{2}O_{3}$ solution, the overall monomer conversion was determined.

The formula for calculating the conversion is as follows:$$H\;\left(\%\right)=\frac{C-\frac{1}{2}\frac{\left(V_{0}-V\right).N}{V_{i}}}{C}$$
where *C* is concentration of the initial monomer; *N* is the concentration of Na_2_S_2_O_3_ solution (N); *V*_0_ is the volume of Na_2_S_2_O_3_ consumed for the blank sample (mL); *V* is the volume of Na_2_S_2_O_3_ consumed to titrate the residual monomers in the sample at time *i* (mL); and $V_{i}$ is the volume of the reaction mixture sample at time *i* (mL). Measurements for calculating the conversion were performed in triplicate. The titration method was used to provide an experimental confirmation of conversion so that we could use these experimental data for later development of models regarding conversion.

Dissolved CPAM was dried in 200 g of distilled water. The viscosity of the polymers was measured by using an Ubbelohde viscometer; the molecular weight of the polymers was estimated from viscosity by using the Mark–Houwink–Sakurada equation:
*η* = *K* × *M*^*α*^(1)
where *η* is the characteristic viscosity of the polymer, *M* is the molecular weight of the polymer chain, and *K* and *α* are constants depending on the nature of the polymer and the solvent [33].

### 2.4. Cationic Polyacrylamide Molecular Weight Analysis

The molecular weights and molecular weight distribution of CPAM were confirmed by gel permeation chromatography (GPC) (detector: RID A, refractive index signal).

### 2.5. Analysis of the Particle Size Distribution of CPAM

The particle size distribution of cationic polyacrylamide nanoparticles was determined by dynamic light scattering (DLS) using the Horiba SZ–100 nanoparticle size measuring device.

### 2.6. Structural Analysis

FTIR spectra were obtained using an IRAffinity–1S Fourier–transform infrared spectrometer (Shimadzu, Japan). ^1^H–NMR spectra were recorded in D_2_O media at room temperature using a Bruker Avance Neo 600 MHz spectrometer.

### 2.7. Statistical Analysis

Experimental data were processed using the statistical software Design–Expert 11.1 (Stat–Ease, MN, USA). The application of experimental design as a powerful statistical tool allowed us to reduce the process variability, combined with the requirement of fewer resources (e.g., time, experimental work); meanwhile, response surface methodology (RSM) allowed us to solve multivariable equations and evaluate the relative significance of several influential factors even in the presence of complex interactions [30].

## 3. Results and Discussion

### 3.1. Characterization of the CPAM

The main infrared absorption bands of the CPAM and the assignments are shown in Figure 4. The bands with frequencies of 3416 cm^−1^ and 1661 cm^−1^ were assigned to stretching vibration of –NH_2_ and C=O, respectively, in the amide groups [21]. The asymmetric adsorption peak at 2926 cm^−1^ was for –CH_3_ and –CH2– [34]. The adsorption peak at 1454 cm^−1^ was for –CH2– flexural vibrations in –CH2–N^+^ [35]. The peak located at 1125 cm^−1^ was attributed to the stretching vibration of C–O from the ester base. The 965 cm^−1^ characteristic adsorption peak was for quaternary ammonium groups. The infrared spectroscopy indicated that the two monomers, AM and DMC, were copolymerized.

Figure 5 displays the ^1^H–NMR spectra of CPAM. The chemical shift of CPAM at about δ_H_ = 0.8545 ppm was ascribed to the protons of –CH_3_– (H_a_). The asymmetric peaks of CPAM at δ_H_ = 1.615 ppm and δ_H_ = 2.118 ppm were attributed to the protons of the backbone methylene and methine groups –CH2– (H_b_) and –CH– (H_c_), respectively. The sharp peak of CPAM at δ_H_ = 3.184 ppm was assigned to the protons of –N^+^(CH_3_)_3_ (H_d_). A peak at δ_H_ = 4.027 ppm was assigned to H_e_ of the O=C–O–CH_2_^+^. The sharp peaks at δ_H_ = 4.69 ppm were assigned to the proton of –N^+^CH_2_– (H_f_). Lastly, the chemical shift at about δ_H_ = 5.283 ppm was ascribed to the protons of O=C–NH_2_ (H_g_). Analysis of the ^1^H–NMR spectral data gave comparable results with the provided data [36].

Figure 6 shows the molecular weight distribution of CPAM as determined by GPC. The results showed that the number–average molecular weight (M_n_) and the weight–average molecular weight (M_w_) of CPAM were about 8,518,300 g/mol and 19,035,000 g/mol, respectively. The molecular mass distribution was expressed as a polydispersity index (PDI). (M_w_/M_n_) = 2.23.

The results of the particle size distribution of the cationic polyacrylamide nanoparticles showed that the average diameter of the polymer particles was 32.2 nm, while the particle size distribution was from 25 to 200 nm (Figure 7). The average diameter and polydispersity index (PI) of the fractionated particle size distribution were measured with a laser instrument under a scattering angle of 173° at an ambient temperature of 25 °C.

Characterization by ^1^H–NMR, FTIR, GPC, and DLS confirmed that we successfully synthesized CPAM with the desired molecular weight by using our proposed response surface models.

### 3.2. Optimal Parameters Affecting Polymerization by Response Surface Methodology

The results of the actual trial synthesis are shown in Table 1. Analysis of variance (ANOVA) was used to build and evaluate the compatibility of the achieved model (Table 2). A model was considered statistically significant when (1) the *p*-Values of the models < 0.05; (2) adequate precision was used to orient the design space greater than 4.0; (3) the lack–of–fit value reflecting the discreteness of the data was not statistically significant; and (4) the R^2^ value was greater than 0.8. The quadratic model in this study has a model F-Value of 137.81, implying that the model is significant. The model *p*-Value less than 0.0001 indicates that the model terms are significant. In this case, A, B, C, AB, AC, BC, A^2^, B^2^, and C^2^ are significant model terms. The lack–of–fit *p*-Value of 0.6754 implies that the lack of fit is not statistically significant [37,38].

The equation describing the dependence of molecular weight on factors such as stirring speed, temperature, and reaction time is a quadratic equation, as follows:M_W_ = −2.46 × 10^8^ + 4.32 × 10^4^ × A + 5.12 × 10^6^ × B + 1.22 × 10^7^ × C − 66.1 × AB − 4.03 × 10^2^ × AC − 3.15 × 10^5^ × BC − 7.15A^2^ − 3.89 × 10^4^ × B^2^ − 6.43 × 10^5^ × C^2^(2)
where A is the stirring rate (rpm), B is the reaction temperature (°C), and C is the reaction time (hours).

Statistical significance is a necessary but insufficient requirement for ensuring the data’s accuracy. R^2^ and adequate precision values were computed to ensure a satisfactory fit of the data (Table 3). The R^2^ score for the present model was 0.996, indicating the best fit for the data. Its value also ranged from 0 to 1. The value of R^2^ adjusted for the current model was 0.9888, which also indicates higher accuracy. Adequate precision measures the signal–to–noise ratio. A ratio greater than 4 is desirable [39]; a ratio of 36.57 indicates an adequate signal.

In addition, several other factors were used to evaluate whether or not the model was fully compatible with the experimental results, based on the predicted and actual value plots and graphs of the residuals versus runs models. The data in Figure 8 also show that the model has a good correlation when the points are concentrated in a straight line, and the distribution of the experimental points is random, with the coefficient of variation CV% low at 2.08.

As shown in Table 4 and Table 5, considering the above criteria, the quadratic model in this study satisfied all four criteria with the model *p*-Value < 0.0001, AP = 94.6566, LOF = 0.1381, and R^2^ = 0.9993 indicating a suitable model. The data in Figure 9 also show good correlation between the predicted and experimental values of the conversions.

For given values of each element, the equation in terms of real factors may be used to create predictions about the response, as follows:%H = −2.76 × 10^3^ + 8.43 × 10^−2^ × A + 73.3 × B + 1.45 × 10^2^ × C − 3.00 × 10^−4^ × AB − 3.00 × 10^−3^ × AC − 2.90 × 10^−1^ × BC − 8.21 × 10^−6^ × A^2^ − 5.78 × 10^−1^ × B^2^ − 8.45C^2^(3)
where A is the stirring rate (rpm), B is the reaction temperature (°C), and C is the reaction time (h).

The molecular weight and conversion of the polymer affected by the difference in the independent variables is visualized through a three–dimensional image of the reaction surface plot (Figure 10 and Figure 11). The plots are represented as a function of two factors at a time, keeping the other factors at fixed levels.

The change in stirring speed caused a significant change in the molecular mass of the polymer produced, as seen in Figure 10. The molecular weight of CPAM drastically increased when the stirring speed increased from 2000 to 2400 rpm, and reached a maximum at a stirring speed of 2400–2600 rpm (Figure 10a–d). When the stirring speed exceeded 3000 rpm, the molecular weight of the CPAM decreased (Figure 10a–d). This could be explained as follows: When increasing the stirring speed, the emulsion was mixed evenly, and the monomer droplets were small and evenly dispersed in the oil phase, increasing the contact between monomer molecules and free radicals, and reducing the steric hindrance of newly formed polymers to monomers and free radicals [40]. As the rate of polymerization increased, the circuit developed rapidly. However, when the stirring speed was too large, the emulsion was strongly agitated, and the contact time of the free radicals with the monomer drops was very short, preventing free radicals from diffusing into the monomer droplets, stimulating the reaction. The polymerization was slowed down, leading to excess monomer, and the molecular weight was reduced.

Based on the response surface methodology in Figure 10a,b,e,f, when the reaction temperature increased from 55 °C to 60 °C, the molecular weight of polyacrylamide cations increased rapidly, and reached its maximum at 60–62 °C. When the temperature continued to increase to 65 °C, the molecular weight of the polymer tended to decrease. Higher temperature was responsible for imidization of the amide groups, resulting in breakage of the imide/amide groups and backbone chain scission, thus decreasing the molecular weight [24,40,41,42,43].

Similarly, in Figure 11c–f, when the reaction time increased from 6 to 7 h, the molecular weight of the polymer increased rapidly. From 7 to 7.5 h, this phase mainly developed polymer chains, and the obtained CPAM had the maximum mass. When we further increased the reaction time to 8 h, the polymer tended to decrease, and the CPAM’s molecular weight decreased.

For conversion efficiency, the effects of stirring speed, temperature, and reaction time are shown in Figure 11. The response surface methodology in Figure 11a,b shows the conversion of the reaction at different stirring speeds from 2000 to 3000 rpm. A stirring speed of 2000–2600 rpm led to a slow increase in the conversion. When the temperature increased from 55 to 61 °C, the reaction efficiency increased very quickly. At 61 °C, the reaction efficiency reached its maximum. If the temperature continued to increase to 65 °C, the reaction efficiency decreased. In general, all polymerization reactions are exothermic [44]. The conversion of polymers strongly depends on the reaction temperature, because it determines the half–life of the initiator. The increase in reaction temperature leads to the formation of active centers, and the reaction process is oriented to form large chains. When the reaction temperature is higher than the decomposition temperature of the initiator, large polymer chains are formed that interfere with the interaction between monomers and free radicals, so the conversion attains lower values. Similar to the reaction temperature, it was found that when increasing the reaction time, the efficiency of the reaction increased, and was the highest when the reaction time was 6.5–7.5 h. If the reaction time continued to increase, the conversion efficiency decreased.

From the response surface models, we found that the optimal parameters for the reaction were a stirring speed of about 2400–2600 rpm, reaction temperature of 60–62 °C, and reaction time of about 6.5–7.5 h.

### 3.3. Verifying the Fit of the Model

To check the significance of the regression coefficients and the compatibility of the regression equations, the experiments at the center were performed as shown in Table 6. The verified experiments provided similar results to the results predicted using Design Expert 11 software, with a small error. This result again confirms that the mathematical method is significant and highly effective in studying the influence of factors such as stirring speed, temperature, and reaction time on the molecular weight and conversion of CPAM.

From the verified experiments combined with the models, the optimal conditions for the cationic inverse emulsion synthesis to achieve high average molecular weight and high conversion are as follows: stirring speed of about 2500–2600 rpm, reaction temperature of 60–62 °C, and reaction time of 7 h.

### 3.4. Application of Response Surface Models

Depending on the needs of the customers, CPAM with different molecular weights is required for mass production. Our response surface models can help to find optimal conditions of temperature, stirring speed, and reaction time for the synthesis of desired–molecular–weight CPAM. CPAM with different molecular weights might have different costs of production. For example, wastewater with a high concentration of organic matter would require high–molecular–weight CPAM. In that case, a combination of temperature, reaction time, and stirring speed is required to synthesize high–molecular–weight CPAM. Our models are able to provide that set of parameters more quickly than the traditional approach. As shown in this study, we applied our models to obtain CPAM with an average molecular weight of about 8,518,300 g/mol. On the other hand, wastewater with a low concentration of organic compounds would require low–molecular–weight CPAM. We could apply the models used in this study to find the optimal conditions to produce low–molecular–weight CPAM. This would save time and costs for mass production.

Previous response surface models were used by Kim to study the pretreatment of paper wastewater with derivatives of polyacrylamide, as the flocculant in the coagulation–flocculation process [31]. Quadratic models were used to correlate dose and pH with chemical oxygen demand, total suspended solids, and sludge volume index [31]. Our models in this study could be combined with those previous models for controlling both synthesis and usage of CPAM in wastewater treatment.

Although our models might help in finding the optimal conditions for CPAM synthesis, the models might have limitations in the case of changes in the monomers’ composition, or if a different molecular weight of CPAM is required. In that case, new models developed using response surface methodology would be required.

## 4. Conclusions

In this study, we were successful in using response surface models to study and synthesize CPAM by reverse emulsion copolymerization. Based on our developed models, we found that the optimal synthesis conditions for 95.948% conversion of CPAM with a molecular weight of 7.639.106 Da were 2500–2600 rpm for stirring rate, 60–62 °C for reaction temperature, and 7 h for reaction time. The response surface model gave predicted values of molecular weight and conversion that matched the experimental values provided by the Ubbelohde viscometer and HIP methods. The models in our study offer a time–saving tool for onsite mass production of specific–molecular–weight CPAM.

## Figures and Tables

**Figure 1 polymers-14-02866-f001:**
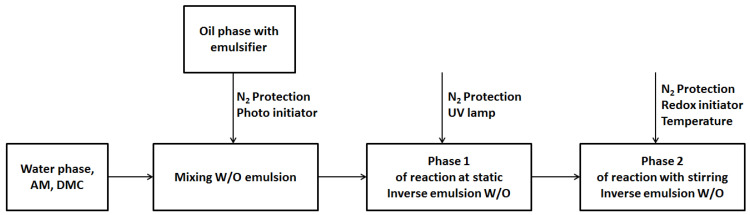
CPAM synthesis process.

**Figure 2 polymers-14-02866-f002:**
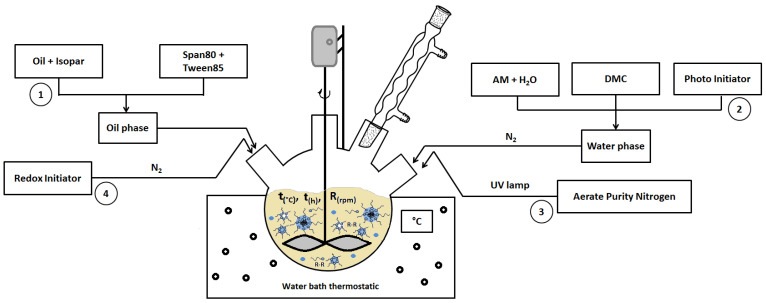
Device diagram.

**Figure 3 polymers-14-02866-f003:**
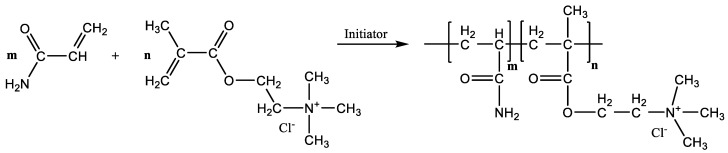
CPAM fusion reaction.

**Figure 4 polymers-14-02866-f004:**
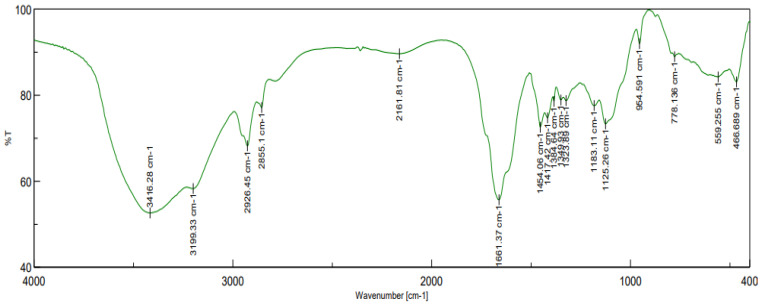
FTIR spectra of CPAM copolymers.

**Figure 5 polymers-14-02866-f005:**
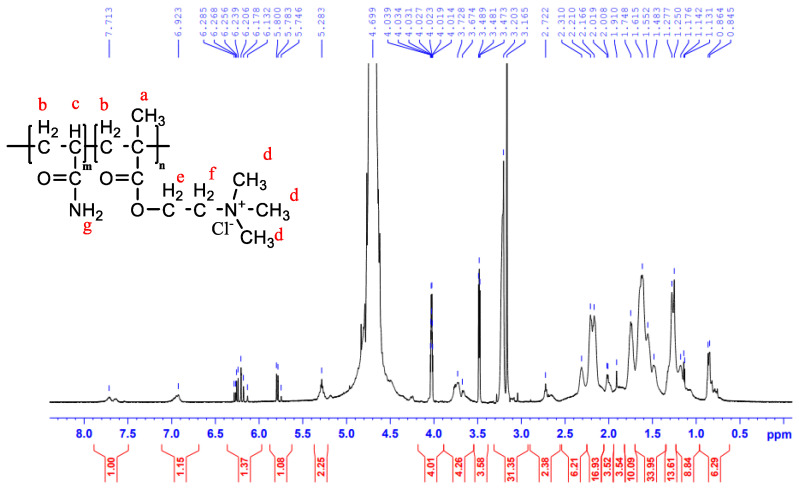
^1^H–NMR spectra of CPAM copolymers.

**Figure 6 polymers-14-02866-f006:**
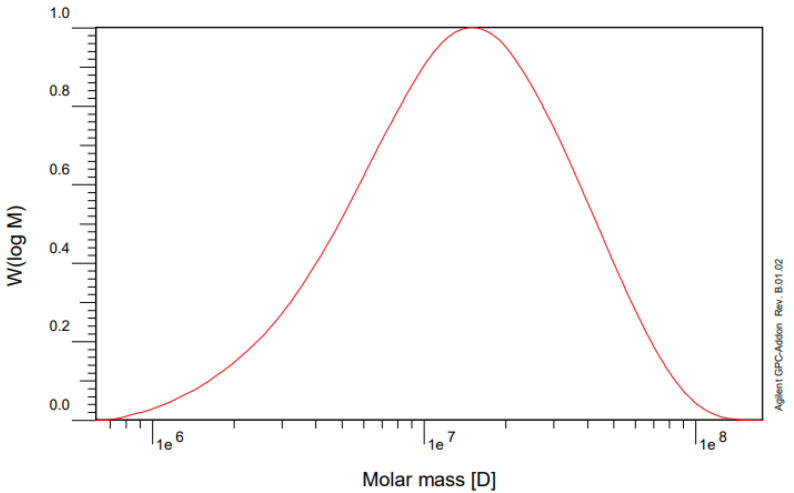
Molecular weight distribution of CPAM determined by GPC (CPAM was synthesized by conditions including stirring speed: 2600 rpm, temperature: 61 °C, and reaction time: 7 h).

**Figure 7 polymers-14-02866-f007:**
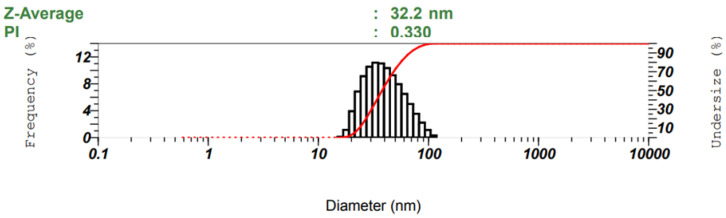
The results of particle size distribution of CPAM by DLS (CPAM was synthesized by conditions including stirring speed: 2600 rpm, temperature: 61 °C, and reaction time: 7 h).

**Figure 8 polymers-14-02866-f008:**
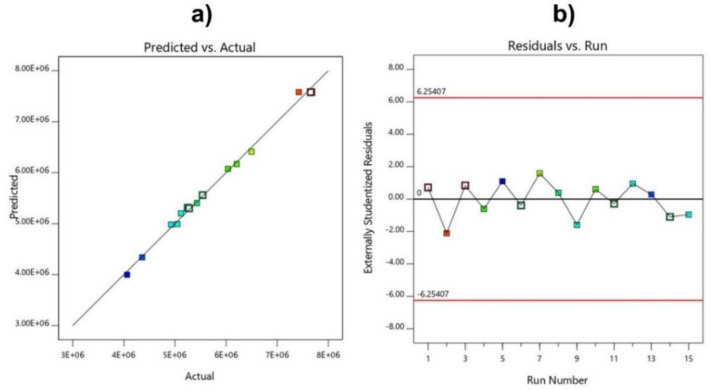
Predicted and actual value plots (**a**), and residuals versus run models (**b**), for molecular weight.

**Figure 9 polymers-14-02866-f009:**
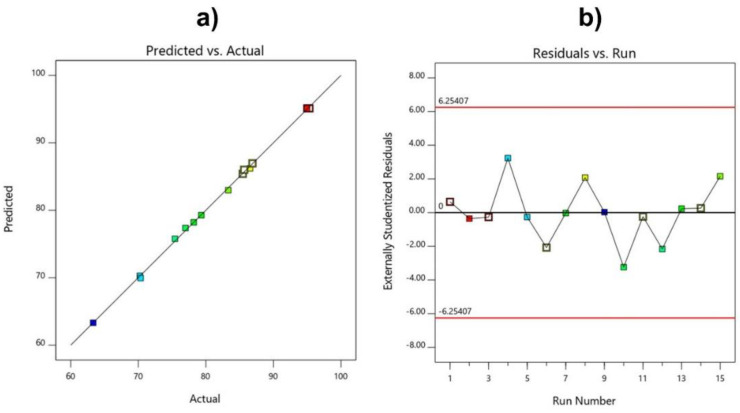
Predicted and actual value plots (**a**), and residuals versus run models (**b**), for conversion.

**Figure 10 polymers-14-02866-f010:**
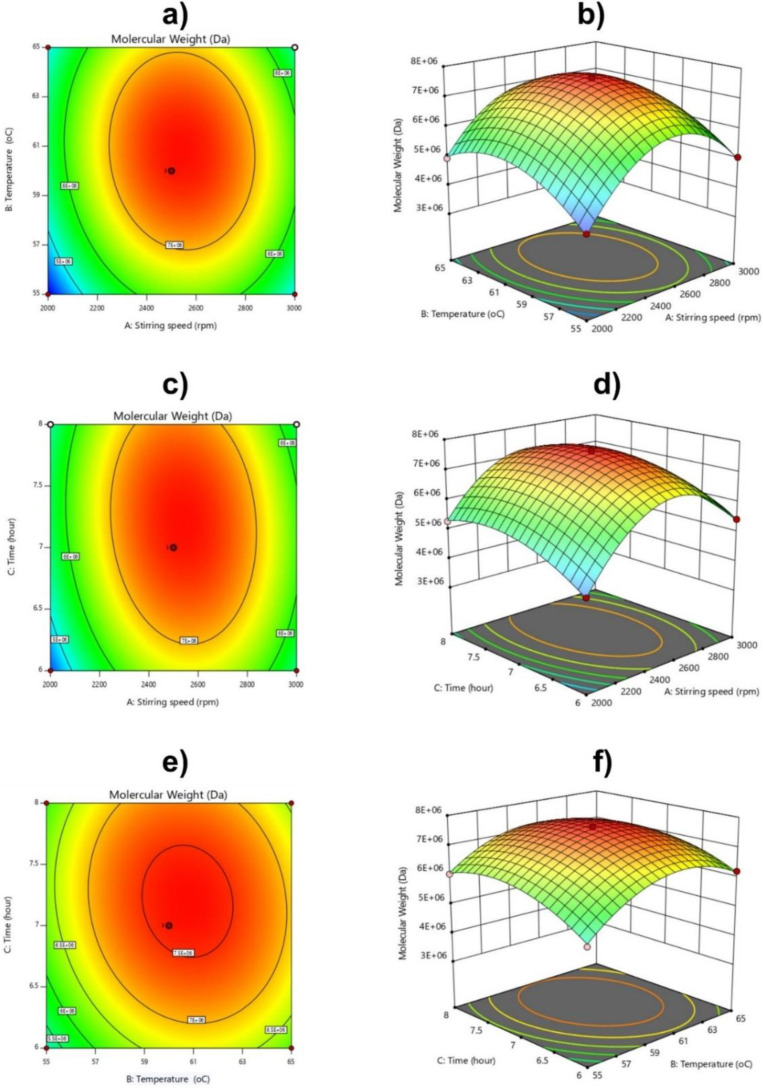
2D contour graphs and 3D response surface: Analysis of interaction effect (**a**,**b**) of stirring speed and temperature, (**c**,**d**) stirring speed and time, and (**e**,**f**) temperature and time on Molecular Weight.

**Figure 11 polymers-14-02866-f011:**
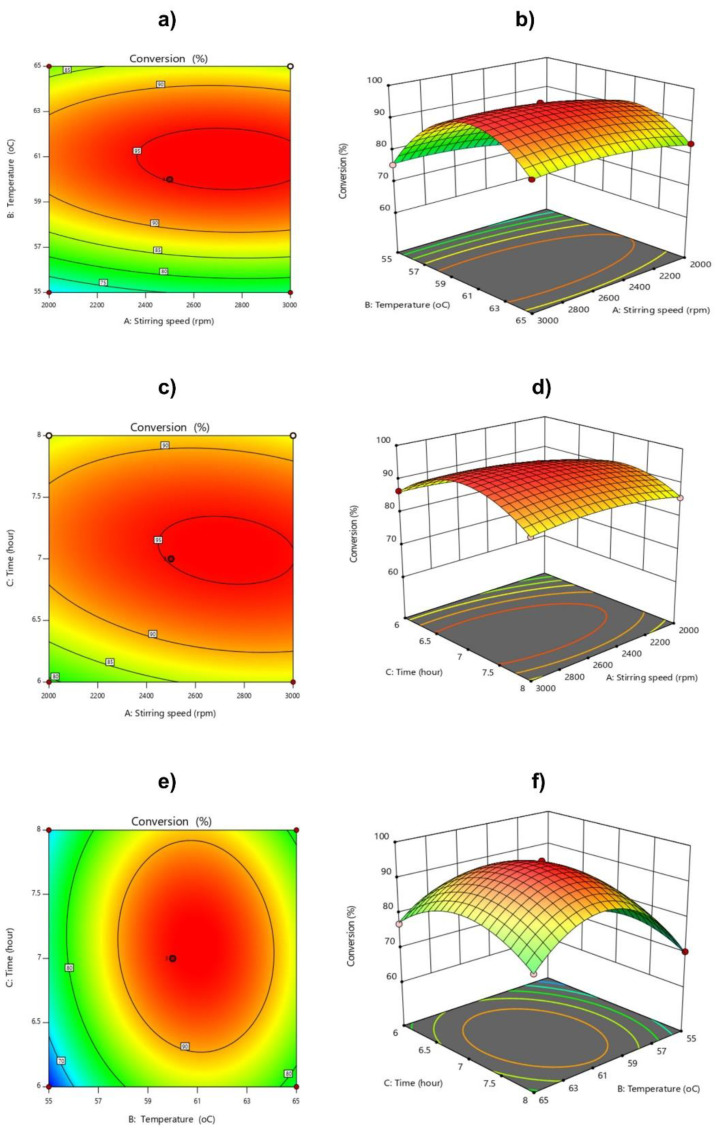
2D contour graphs and 3D response surface: analysis of the interaction effects of stirring speed and temperature (**a**,**b**), stirring speed and time (**c**,**d**), and temperature and time (**e**,**f**) on conversion.

**Table 1 polymers-14-02866-t001:** Results of the actual trial synthesis.

Reaction No.	Stirring Speed (rpm)	Temperature (°C)	Time (h)	Molecular Weight (Da)	Conversion (%)
1	2000	55	7	4.06 × 10^6^	70.23
2	3000	55	7	5.05 × 10^6^	75.43
3	2000	65	7	4.92 × 10^6^	83.32
4	3000	65	7	5.25 × 10^6^	85.46
5	2000	60	6	4.36 × 10^6^	79.32
6	3000	60	6	5.43 × 10^6^	86.54
7	2000	60	8	5.28 × 10^6^	85.67
8	3000	60	8	5.54 × 10^6^	86.90
9	2500	55	6	5.12 × 10^6^	63.32
10	2500	65	6	6.21 × 10^6^	76.98
11	2500	55	8	6.04 × 10^6^	70.34
12	2500	65	8	6.50 × 10^6^	78.21
13	2500	60	7	7.67 × 10^6^	95.01
14	2500	60	7	7.65 × 10^6^	95.34
15	2500	60	7	7.42 × 10^6^	94.98

**Table 2 polymers-14-02866-t002:** ANOVA for the response surface quadratic model (Cor Total: Corrected Total Sum of Squares).

Source	Sum of Squares	Df	Mean Squares	F-Value	*p*-Value	
Model	1.79 × 10^13^	9	1.98 × 10^12^	137.81	<0.0001	Significant
A	8.76 × 10^11^	1	8.76 × 10^11^	60.89	0.0006	
B	8.53 × 10^11^	1	8.53 × 10^11^	59.29	0.0006	
C	6.27 × 10^11^	1	6.27 × 10^11^	43.54	0.0012	
AB	1.09 × 10^11^	1	1.09 × 10^11^	7.59	0.0401	
AC	1.67 × 10^11^	1	1.67 × 10^11^	11.30	0.0201	
BC	9.90 × 10^10^	1	9.90 × 10^10^	6.88	0.0469	
A^2^	1.18 × 10^10^	1	1.18 × 10^13^	819.00	<0.0001	
B^2^	3.50 × 10^10^	1	3.50 × 10^13^	242.92	<0.0001	
C^2^	1.53 × 10^12^	1	1.53 × 10^12^	106.07	0.0001	
Residual	7.19 × 10^10^	5	1.44 × 10^10^			
Lack of Fit	3.40 × 10^10^	3	1.13 × 10^10^	0.5968	0.6754	Not significant
Pure Error	3.80 × 10^10^	2	1.90 × 10^10^			
Cor Total	1.79 × 10^13^	14				

**Table 3 polymers-14-02866-t003:** Results of the analysis of the suitability of the experimental model (CV: coefficient of variation).

Std. Dev.	1.20 × 10^5^	R^2^	0.9960
Mean	5.77 × 10^6^	Adjusted R^2^	0.9888
C.V.%	2.08	Predicted R^2^	0.9649
		Adequate Precision	36.57

**Table 4 polymers-14-02866-t004:** ANOVA for the response surface quadratic model (Cor Total: Corrected Total Sum of Squares).

Source	Sum of Squares	Df	Mean Squares	F-Value	*p*-Value	
Model	1299.42	9	144.38	853.03	<0.0001	Significant
A	31.17	1	31.17	184.13	<0.0001	
B	249.20	1	249.20	1472.35	<0.0001	
C	27.98	1	27.98	165.28	<0.0001	
AB	2.34	1	2.34	13.83	0.0137	
AC	8.97	1	8.97	53.00	0.0008	
BC	8.38	1	8.38	49.52	0.0009	
A^2^	15.55	1	15.55	91.90	0.0002	
B^2^	770.70	1	770.70	4553.46	<0.0001	
C^2^	263.64	1	263.64	1557.65	<0.0001	
Residual	0.8463	5	0.1693			
Lack of Fit	0.7665	3	0.2555	6.40	0.1381	not significant
Pure Error	0.0798	2	0.0399			
Cor Total	1300.26	14				

**Table 5 polymers-14-02866-t005:** Results of the analysis of the suitability of the experimental model.

Std. Dev.	0.4114	R^2^	0.9993
Mean	81.80	Adjusted R^2^	0.9982
C.V. %	0.5029	Predicted R^2^	0.9904
		Adequate Precision	94.6566

**Table 6 polymers-14-02866-t006:** Molecular weight and conversion of CPAM according to verified experiments.

No.	Stirring Speed (rpm)	Temperature (°C)	Reaction Time (h)	Verified Experiment		Predicted	
				M_w_ (Da)	Conversion (%)	M_w_ (Da)	Conversion (%)
1	2400	60	7	7,325,663	94.99	7,475,073	95.36
2	2400	61	7	7,357,949	95.47	7,508,111	95.94
3	2400	62	7	7,315,272	95.16	7,463,313	95.37
4	2500	60	7	7,480,452	95.07	7,612,678	95.86
5	2500	61	7	7,486,342	95.65	7,639,107	96.42
6	2500	62	7	7,445,982	95.05	7,587,700	95.82
7	2600	60	7	7,465,625	95.92	7,607,362	96.21
8	2600	61	7	7,474,824	95.96	7,627,182	96.73
9	2600	62	7	7,437,792	95.33	7,569,166	96.10
10	2400	60	6.5	7,002,888	90.43	7,152,600	92.17
11	2400	61	6.5	7,052,313	91.92	7,201,373	92.89
12	2400	62	6.5	7,028,117	91.57	7,172,309	92.47
13	2500	60	6.5	7,163,192	92.08	7,310,367	92.82
14	2500	61	6.5	7,207,192	92.67	7,352,530	93.52
15	2500	62	6.5	7,170,218	91.78	7,316,858	93.06
16	2600	60	6.5	7,173,829	92.14	7,325,213	93.32
17	2600	61	6.5	7,217,162	93.01	7,360,767	93.98
18	2600	62	6.5	7,171,252	92.75	7,318,486	93.50
19	2400	60	7.5	7,326,574	92.57	7,476,096	94.32
20	2400	61	7.5	7,343,532	93.87	7,493,400	94.76
21	2400	62	7.5	7,284,210	93.29	7,432,867	94.04
22	2500	60	7.5	7,342,719	93.33	7,593,539	94.68
23	2500	61	7.5	7,358,101	92.33	7,604,233	95.09
24	2500	62	7.5	7,386,350	92.99	7,537,092	94.34
25	2600	60	7.5	7,386,701	93.20	7,568,062	94.87
26	2600	61	7.5	7,390,704	94.11	7,572,147	95.25
27	2600	62	7.5	7,348,429	93.82	7,498,397	94.47

## Data Availability

Not applicable.

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
