# Peer review of "Optimizing the Conditions of Cationic Polyacrylamide Inverse Emulsion Synthesis Reaction to Obtain High–Molecular–Weight Polymers"

_polymers, 2022, doi:10.3390/polym14142866_

Round 1

Reviewer 1 Report

Dear Author,

The author creates a model to see the effect of temperature, stirring speed, and molecular weight on the polymerization of cationic acrylamide. It's an interesting read but needs significant improvement. Here are my comments:

(1) Introduction is too short and doesn't explain the complete background and why this study is needed.

(2) Why this type of study is needed and how this is economically viable. The author failed to tell the readers the novelty and need for this work and what knowledge we will gain.

(3) The author should present the synthesis including characterization such as NMR, FTIR, and molecular weight first in the result discussion followed by statistical analysis. They should also be able to explain why and what is the relation between increasing molecular weight with stirring speed and temperature.

(4) Why do we need the titration experiment when from the molecular weight we can find the yield. 

Please address this issues along with complete background and literature on doing this study

Author Response

We appreciate valuable comments from the reviewers. Following their comments, we have thoroughly revised our manuscript to improve its clarity. The modifications that we made were incorporated as red-colored texts in the revised manuscript. We also present our response to each comment of the reviewers as in the attached file below.

Reviewer 2 Report

This paper mainly focused on the experimental design method for cation inverse emulsion synthesis of polyacrylamide and built a response surface model for predicting average molecular weight and the reaction yield based on influence factors such as temperature, reaction time, and stirring speed. Although the use of experimental design to find out the model of dependence for a reaction results is new in this particular reaction, I don’t find it is a significant advancement in the polymer synthesis field, especially for the synthesis of polyacrylamide. In addition, there are several papers already described the optimization of cation inverse emulsion synthesis of polyacrylamide experimentally. In the paper, the author was unable to properly describe what’s new they are bringing in this. This work would be interesting if the author shows some serious application of their method or their materials. I think this paper is not up to the level of novelty and significance of the Polymers. To make it to the level of Polymers the author needs to show some applications and re-write the discussion part. With that, there is a lot of typo error throughout the manuscript, that needs correction.

Author Response

(The authors gave the same response as above.)

Round 2

Reviewer 1 Report

Manuscript is ready for publication

Reviewer 2 Report

The author has impressed me with their work on the revision. They have changed the manuscript as suggested by the reviewers. The manuscript is now acceptable in its present form.